# ViewPCGC: View-Guided Learned Point Cloud Geometry Compression

## ABSTRACT

With the rise of immersive media applications such as digital museums, virtual reality, and interactive exhibitions, point clouds, as a three-dimensional data storage format, have gained increasingly widespread attention. The massive data volume of point clouds imposes extremely high requirements on transmission bandwidth in the above applications, gradually becoming a bottleneck for immersive media applications. Although existing learning-based point cloud compression methods have achieved specific successes in compression efficiency by mining the spatial redundancy of their local structural features, these methods often overlook the intrinsic connections between point cloud data and other modality data (such as image modality), thereby limiting further improvements in compression efficiency. To address the limitation, we innovatively propose a view-guided learned point cloud geometry compression scheme, namely ViewPCGC. We adopt a novel self-attention mechanism and cross-modality attention mechanism based on sparse convolution to align the modality features of the point cloud and the view image, removing view redundancy through Modality Redundancy Removal Module (MRRM). Simultaneously, side information of the view image is introduced into the Conditional Checkboard Entropy Model (CCEM), significantly enhancing the accuracy of the probability density function estimation for point cloud geometry. In addition, we design a View-Guided Quality Enhancement Module (VG-QEM) in the decoder, utilizing the contour information of the point cloud in the view image to supplement reconstruction details. The superior experimental performance demonstrates the effectiveness of our method. Compared to the state-of-the-art point cloud geometry compression methods, ViewPCGC exhibits an average performance gain exceeding 10% on D1-PSNR metric.

## CCS CONCEPTS

• **Theory of computation** → **Data compression**.

## KEYWORDS

Point Cloud, Geometry Compression, Multimodal Learning, Deep Learning

## 1 INTRODUCTION

As 3D data acquisition technology progresses swiftly, point clouds have emerged as a crucial format for 3D data. They are widely used

*ACM MM, 2024, Melbourne, Australia*

© 2024 Copyright held by the owner/author(s). Publication rights licensed to ACM.
ACM ISBN 978-x-xxxx-xxxx-x/YY/MM
https://doi.org/10.1145/nnnnnnn.nnnnnnn

in various fields such as digital museums [3, 41], virtual reality [1, 12], 3D reconstruction [7, 26, 50], cultural heritage preservation [31, 39], and medical imaging [38, 42]. In contrast to 2D data (such as images), point clouds have become indispensable due to their ability to describe objects in the physical world more accurately. However, high-quality point cloud data often comes with a huge volume of data, posing significant challenges to storage and transmission. Therefore, efficient point cloud geometry compression technology is particularly important.

In fact, point cloud geometry compression methods have developed rapidly in the past decade. Before 2020, research in the field primarily focused on optimizing the MPEG standardized point cloud compression method G-PCC [15]. G-PCC is well known for its excellent compatibility and standardization across different application scenarios and platforms. Nonetheless, the advent of learning-based point cloud compression techniques has amplified the limitations of G-PCC concerning the performance of point cloud geometry compression. Researchers' attention has shifted towards deep learning-based approaches. In the early stages, numerous point-based methods [14, 17, 19, 48] are developed for object point clouds. These methods perform excellently when dealing with simple structures and fewer points, such as ShapeNet [5], but their performance degrades with dense point clouds due to the massive number of points and poor compression effects. Subsequently, researchers design voxel-based methods [29, 30, 45] targeting the spatial structure of dense point clouds. Point clouds are discretely depicted as three-dimensional grids, enabling more efficient storage and processing of point cloud data. However, the extensive spatial scale of dense point clouds results in many voxels being unoccupied, leading to spatial wastage and significantly affecting computational efficiency. In order to address the shortcoming, some studies [43, 44, 47] reduce the computational complexity by employing sparse convolution [9], which significantly improved the compression efficiency of learning-based models in dense point cloud geometry.

However, the above learning-based models usually extract features from the intrinsic geometric structure of the point cloud, reducing spatial redundancy through successive downsampling. Although they effectively utilize the intrinsic attributes of point clouds, they often overlook the potential insights gained from external views. Compared to point clouds, the bit cost of the view image during transmission is almost negligible. Such view image can provide additional contextual information, which is not easily visible from the point clouds alone, leading to more efficient compression. In light of this, we propose View-Guided Learned Point Cloud Geometry Compression (ViewPCGC) in this paper. The core of this framework is the use of view information to assist in point cloud compression. By leveraging self-attention and cross-attention mechanisms based on sparse convolution, we designed Modal Redundancy Reduction Module (MRRM) to eliminate the

redundancy between modalities. In the entropy model, we adopt a conditional entropy model, introducing side information of the view hyperprior as context for point cloud encoding, making the probability estimation of entropy parameters more accurate. At the end of the decoder, to fully utilize the view information, we design a quality enhancement network that further uses the structural information of the view to restore local details of the point cloud.

The contributions of our proposed method can be summarized as follows:

- We propose a View-Guided Learned Point Cloud Geometry Compression (ViewPCGC) framework by utilizing view information. To the best of our knowledge, it is the first work to leverage modality information from the projected view in deep learning-based object point cloud geometry compression.
- Modality Redundancy Removal Module (MRRM) is crafted to eliminate redundancy across modalities, aiming for an enhanced compression ratio. Specifically, it harnesses both self-attention and cross-attention mechanisms, which are built upon sparse convolution, to facilitate a more effective fusion of information across channels and modalities.
- Conditional Checkboard Entropy Model (CCEM) is engineered to uncover the interdependencies among modalities. It employs the side information from the latent representation of view images as a prior for estimating the probability of entropy parameters, which significantly enhances the compression efficiency.
- View-Guided Quality Enhancement Module (VG-QEM) is designed to leverage the structural information of the view for the purpose of restoring local details in the point cloud, which significantly preserves the edge features and geometric integrity of the point cloud, maintaining its detailed structure and quality.
- Experimental results demonstrate that ViewPCGC achieves state-of-the-art performance when compared to existing point cloud geometry compression methods.

## 2 RELATED WORK

### 2.1 Learning-based Point Cloud Compression

Over recent years, the field of deep learning has seen significant growth, leading to an increased number of learning-based methods in point cloud geometry compression. According to the data structure, the leading point cloud geometry compression frameworks can be classified into three main types: point-based methods, octree-based methods, and voxel-based methods. Point-based methods draw inspiration from works like PointNet [33] and PointNet++ [34], utilizing 3D convolutional networks to primarily extract features from point clouds and restore them at the decoding end. Yan et al. [48] introduce the first learning-based point cloud compression framework, defining a series of optimization paradigms for this field. Gao et al. [14] employ graph convolutional methods to construct local graphs for the neighbors of each point and associate attributes, performing well on the ShapeNetCorev2 dataset [5]. He et al. [17] focus more on modeling local representations, learning three types of feature embeddings to encode local geometric features effectively, thus efficiently solving the point clustering

problem. The research on point-based methods primarily targets datasets with fewer point cloud points, such as ShapeNet [5]. With the expansion of the point cloud size, the temporal and spatial complexity of these methods escalates rapidly. Octree-based methods address the spatial sparsity of lidar point clouds by recursively partitioning the three-dimensional space into progressively smaller regions to organize the point cloud data. Huang et al. [18] pioneer the first octree-based learned point cloud compression framework, devising an entropy model based on octree to forecast symbol probabilities. Fu et al. [13] apply a large-scale transformer structure, utilizing the prior information of sibling and ancestor nodes, which enhances the probability prediction capability of the entropy model and achieves significant performance breakthroughs. Song et al. [40] design grouped contexts, which greatly accelerate the encoding and decoding time of the model without sacrificing performance. Voxel-based methods encode continuous three-dimensional space into regular, discrete grids, demonstrating strong adaptability for dense object point clouds that occupy a smaller spatial scale but have a larger number of points. PCGCv1 [45] is a representative work in the field, which voxelizes the point cloud geometry and feeds it into a variational autoencoder for sampling. It replaces the traditional MSE loss with a binary cross-entropy loss based on occupancy grids. Building on PCGCv1, PCGCv2 [44] employs sparse convolution to further eliminate the vast spatial redundancy of point clouds, achieving a compact representation and significantly improving Rate-Distortion (RD) performance. SparsePCGC [43] designs multiple groups context strategies, further enhancing encoding performance. Despite the emergence of numerous efficient point cloud geometry compression methods, these methods are all striving to harness the spatial redundancy inherent in point clouds, overlooking the gains from other modality information on point clouds, which also makes performance improvements in recent point cloud geometry compression work challenging.

### 2.2 Multimodal Compression

Multimodal compression aims to enhance data compression efficiency by fusing diverse data types. The core advantage of the method lies in its ability to leverage the complementary information among different modalities, thus achieving a higher compression ratio than single-modality compression. To date, image compression has witnessed the emergence of an extensive array of multimodal compression schemes across its various subfields. In stereo image compression, Liu et al. [24] are the first to propose a learning-based stereo image compression network that uses a parameter skip function to remove modality redundancy between two views. Although the performance improvement is not significant, it lays the foundation for subsequent research. Deng et al. [10] utilize a homography matrix to apply a transformation from the left view to the right view and obtained performance gains based on residual encoding. Zhang et al. [51] propose CAMSIC, introducing a novel content-aware masking image modeling technique and adopting a transformer-based entropy model to capture spatial and disparity dependencies, achieving a performance breakthrough.

In RGB-X image compression, Chen et al. [8] use a prior fusion method to extract multiscale information from the encoder and decoder, fusing cross-modality features to assist in the generation

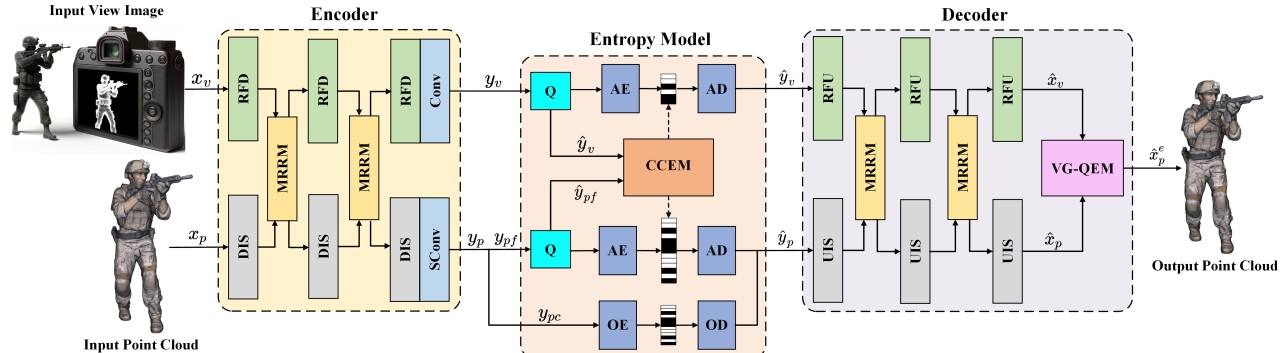

**Figure 1: The overall network architecture of ViewPCGC. "DIS" denotes a composite module that encompasses downsampling, an Inception-Residual Network (inherited from [44]), and a sparse self-attention (SSA) module. Conversely, the "UIS" module parallels "DIS" in structure but diverges by substituting downsampling with upsampling processes. "MRRM" is an acronym for Modality Redundancy Removal Module. The abbreviations "Conv" and "SConv" refer to 2D convolution for image data and 3D sparse convolution for point cloud data, respectively. "CCEM" represents Conditional Checkerboard Entropy Model. "RFD" and "RFU" are acronyms for residual feature downsampling and upsampling. "VG-QEM" denotes View-Guided Quality Enhancement Module. "Q" denotes Quantization. "AE" and "AD" are arithmetic encoder and decoder, while "OE" and "OD" are octree encoder and decoder.**

of depth maps. Lu *et al.* [25] design a multimodal compression framework that uses learnable parameters to perform affine transformations, converting the latent features of infrared features into RGB features and aggregating the features to achieve efficient multimodal compression. Zheng *et al.* [52] introduce intra-modality attention and cross-modality attention, effectively eliminating redundancy between modalities and significantly enhancing compression performance. In text-guided image compression, Jiang *et al.* [21] address issues such as blurriness and loss of detail at low bitrates by using text as semantic prior information to guide image compression, significantly enhancing compression quality at low bitrates. Qin *et al.* [35] leverage the CLIP text encoder, integrating text-level semantic details into image decoding to enhance the compression process.

In the field of point cloud geometry compression, attention is increasingly attracted towards the utilization of view information. Lin *et al.* [23] introduce a pioneering multimodal compression framework specifically designed for lidar point clouds. This innovative approach leverages depth estimation techniques to reconstruct 3D scene information from 2D images. The reconstructed scene information is then systematically aligned with different levels within an octree structure, aiding in the accurate estimation of symbol probability densities.

From the analysis above, we can infer that current multimodal compression works are primarily concentrated in the image compression domain, with multimodal compression targeting point clouds being exceedingly rare. This scarcity is attributed to the highly sparse and irregular characteristics of point cloud data, which make it challenging to effectively extract and utilize high-level abstract semantic features, thereby limiting the ability to align information with other modalities. The work proposed by Lin *et al.* [23] presents one of the few multimodal compression endeavors involving point clouds and images. However, this work fails to adequately address modal redundancy, which in turn limits the

potential for notable performance improvements. Therefore, in the current field of point cloud geometric compression, particularly for dense object geometries, there is an urgent need for targeted multimodal compression works.

## 3 METHODOLOGY

### 3.1 Overview

The overall architecture of ViewPCGC is depicted in Fig. 1. The network is innovatively designed with a dual-branch structure. It integrates a framework based on sparse convolution to compress point cloud data, while utilizing a local attention module to extract features from images. The original point cloud $x_p$ and the corresponding view image $x_v$ are fed into the encoder. Here, $x_p$ is subjected to downsampling and feature aggregation through DIS, while $x_v$ undergoes downsampling and residual feature extraction via RFD. Subsequently, the latent representations of both modalities are fed into the MRRM for modality interaction, effectively reducing the redundancy in the point cloud modality. Following the encoding process, we obtain the latent representations for the point cloud, $y_p$, and the view image, $y_v$. The $y_p$ is then divided into the coordinate component, $y_{pc}$, and the feature component, $y_{pf}$, to facilitate further compression. Both $y_{pf}$ and $y_v$ are directed to the quantizers, while $y_{pc}$ undergoes octree-based compression through OE and OD. The quantized latent representations for the point cloud features, $\hat{y}_{pf}$, and the image, $\hat{y}_v$, are then input into CCEM for precise symbol probability estimation. On the decoder side, $\hat{y}_{pf}$ and $y_{pc}$ are concatenated to obtain $\hat{y}_p$, which is used for point cloud feature reconstruction. Both $\hat{y}_p$ and $\hat{y}_v$ are then processed through UIS and RFU, respectively, for upsampling and feature restoration. Finally, the VG-QEM is utilized to enhance the geometry structure and finer details of the point cloud further, thereby yielding the reconstructed point cloud $\hat{x}_p^e$. We designate the encoder, split operation, quantizer, concat operation, and decoder

with the symbols $E(\cdot)$, $Split(\cdot)$, $Q(\cdot)$, $Cat(\cdot)$, and $D(\cdot)$, respectively. The primary inference process, excluding the entropy model, is

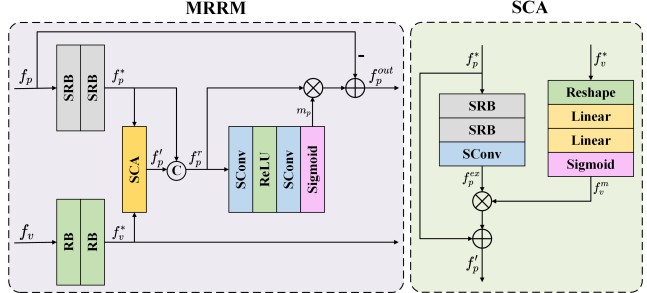

**Figure 2: The architecture of MRRM and sparse cross-attention (SCA). "Reshape" represents the mapping of image features to two-dimensional tensors matching the point cloud feature dimensions, and "Linear" stands for the fully connected layer. "RB" indicates the residual block. "SRB" represents the sparse residual block. "ReLU" stands for the Rectified Linear Unit. "Sigmoid" is the Sigmoid function.**

succinctly represented as follows:

$$
\begin{aligned}
y_p, y_v &= E(x_p, x_v), \\
y_{pc}, y_{pf} &= Split(y_p), \\
\hat{y}_{pf}, \hat{y}_v &= Q(y_{pf}, y_v), \\
\hat{y}_p &= Cat(y_{pc}, \hat{y}_{pf}), \\
\hat{x}_p^e &= D(\hat{y}_p, \hat{y}_v).
\end{aligned}
\tag{1}
$$

To clearly describe the details of each module, we illustrate the detailed structures of certain modules (including DIS, UIS, RFD, RFU, SSA) in the Supplementary Materials. It should be emphasized that due to the massive volume of point cloud data, we adopt linear residual mechanism in place of non-local attention mechanism [46], which significantly improves the inference speed while essentially maintaining consistent performance.

### 3.2 Modality Redundancy Removal Module

In the encoder and decoder stages, we utilize the MRRM to eliminate modal redundancy. The framework of the MRRM is shown in Fig. 2. MRRM contains two branches, each receiving the input latent representations of the point cloud $f_p$ and the image $f_v$, respectively. For point cloud processing, SRB is used to obtain aggregated point cloud features $f_p^*$, while RB is employed on the image side to attain extracted image features $f_v^*$. Subsequently, the features extracted from both are input into sparse cross-attention (SCA) to get the output point cloud latent representation $f_p'$. Similar to sparse self-attention (SSA), we adopt the residual mechanism instead of scale dot-product attention to accelerate the inference process. Within SCA, the latent representation of the image is reshaped into a two-dimensional tensor $f_v^s$ that resembles the sparse tensor feature structure. Through continuous linear layer transformations, channel dimension alignment is achieved. The Sigmoid function maps image features to a mask $f_v^m$ with values ranging from $[0, 1]$, which is then used to weight the extracted point cloud latent representation $f_p^{ex}$. $f_p^{ex}$ is obtained from $f_p^*$ through consecutive SRBs and

SConv. The final output is the point cloud latent representation $f_p'$, which have undergone modality interaction. The processing of SCA can be summarized as:

$$
\begin{aligned}
f_p' &= f_p^{ex} * f_v^m + f_p^* \\
\text{with} \quad f_v^m &= \sigma\left(F_L\left(f_v^s\right)\right),
\end{aligned}
\tag{2}
$$

where $\sigma(\cdot)$ represents the Sigmoid function, and $F_L(\cdot)$ encompass the continuous linear layers. After the application of SCA, we concatenate $f_p'$ with $f_p^*$ along the channel dimension. The concatenation facilitates the feature fusion, resulting in a refined latent representation denoted as $f_p^r$. Subsequently, the latent mask $m_p$ is derived by mapping $f_p^r$ to the weight domain, which serves to selectively emphasize certain region within the point cloud latent representation. Finally, we remove the redundant information from the input point cloud feature $f_p$. The above processing procedure of MRRM can be formulated as:

$$
\begin{aligned}
f_p^{out} &= f_p - f_p^r * m_p \\
\text{with} \quad m_p &= \sigma\left(F_p\left(f_p^r\right)\right),
\end{aligned}
\tag{3}
$$

where $F_p(\cdot)$ is composed of two consecutive sparse convolution layers. After a thorough examination of the various components of MRRM and their interactions, it is evident that MRRM enhances the expressive capability of point cloud features through effective feature fusion and the elimination of redundant modality information, which is particularly crucial in complex point cloud geometry compression tasks.

### 3.3 Conditional Checkboard Entropy Model

Previous point cloud geometry compression methods typically model the latent representation directly [43, 44] or employ basic side information in the hyperprior network to aid probability prediction [36]. Such methods, however, encounter limitations in acquiring sufficient prior information, which in turn impacts the capacity to to predict symbol probabilities accurately. VoxelDNN [29] aims to address these limitations by employing an autoregressive context prediction method similar to those utilized in image compression [27], effectively harnessing spatial information from the adjacent neighborhood. Despite this advancement, the method suffers from exceedingly prolonged decoding time, rendering it impractical for direct model deployment. In response to these challenges, we introduce the Conditional Checkboard Entropy Model (CCEM), which is inspired by [16]. The architecture of CCEM is depicted in Fig. 3.

After the encoding phase, $\hat{y}_{pf}$ and $\hat{y}_v$ are directed towards the hyper encoder, to derive hyperprior latent representations, culminating in $\hat{z}_{pf}$ and $\hat{z}_v$. The hyperprior latent representations are fed into the hyper decoder, leading to the extraction of spatial distribution information $\varphi_p$ and $\varphi_v$, which are divided into anchor parts $\varphi_p^a$, $\varphi_v^a$ and non-anchor parts $\varphi_p^n$, $\varphi_v^n$, respectively. It should be emphasized that CCEM aims to refine the accuracy of probability density estimations for point cloud symbols. For images, due to their limited transmission overhead, after obtaining $\varphi_v$, we directly feed $\varphi_v$ into the Entropy Prediction Module (EPM) to acquire entropy parameters of Gaussian distribution $\mu_v$ and $\sigma_v^2$. The quantized

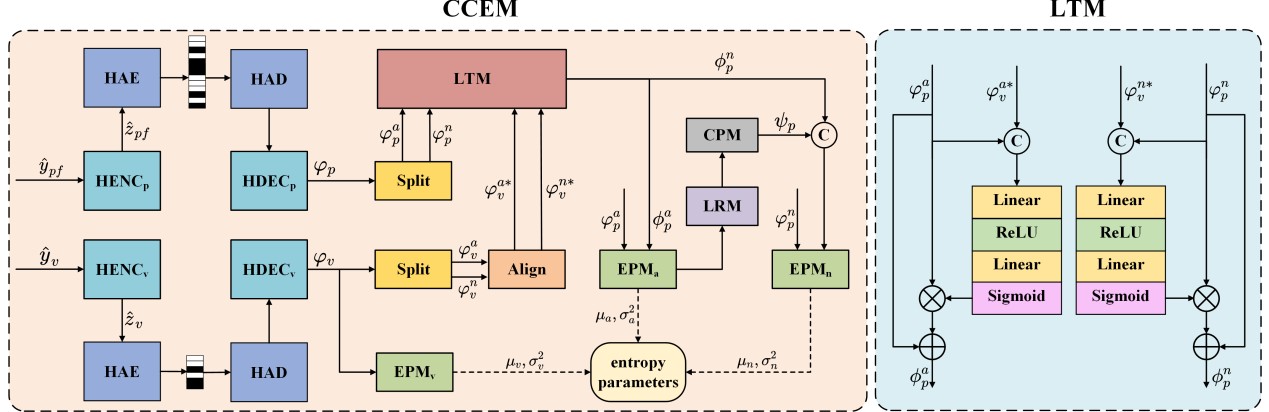

**Figure 3: The architecture of CCEM and Latent Transform Module (LTM). "HENC" and "HDEC" denote hyper encoder and decoder, respectively. "HAE" and "HAD" refer to hyper arithmetic encoder and decoder. "LRM", "CPM", and "EPM" stand for Latent Recovery Module, Context Prediction Module, and Entropy Prediction Module, respectively.**

view image latent representation $\hat{y}_v$ is characterized as follows:

$$p_{\hat{y}_v|\varphi_v}(\hat{y}_v \mid \varphi_v) \sim \mathcal{N}\left(\mu_v, \sigma_v^2\right)$$
$$\text{with} \quad \mu_v, \sigma_v^2 = g_{ep}^v(\varphi_v), \tag{4}$$

where $\mathcal{N}(\cdot)$ denotes Gaussian distribution function, and $g_{ep}^v(\cdot)$ stands for EPM$_v$. To facilitate a more effective interaction with the side information of point cloud, $\varphi_v^a, \varphi_v^n$ are meticulously aligned with $\varphi_p^a, \varphi_p^n$, respectively, resulting in $\varphi_v^{a*}$ and $\varphi_v^{n*}$. All anchor and non-anchor latent representations are fed into Latent Transform Module (LTM) to achieve modality fusion. The fused anchor latent representation $\phi_p^a$ is subsequently forwarded to EPM$_a$ to facilitate the estimation of the likelihood of the anchor point cloud latent representation. The quantized anchor point cloud latent representation $\hat{y}_{pf}^a$ are derived as follows:

$$p_{\hat{y}_{pf}^a|\varphi_p^a,\phi_p^a}\left(\hat{y}_{pf}^a \mid \varphi_p^a, \phi_p^a\right) \sim \mathcal{N}\left(\mu_a, \sigma_a^2\right)$$
$$\text{with} \quad \mu_a, \sigma_a^2 = g_{ep}^a(\varphi_p^a, \phi_p^a), \tag{5}$$

where $g_{ep}^a(\cdot)$ represents EPM$_a$. Alternatively, $\hat{y}_{pf}^a$ is recovered from $\mu_a$ and $\sigma_a^2$ in Latent Recovery Module (LRM). Context Prediction Module (CPM), employing sparse mask convolution (evolved from [16]), predicts the local context $\psi_p$ leveraging the neighbor information of $\hat{y}_{pf}^a$. Consequently, the quantized non-anchor point cloud latent representation $\hat{y}_{pf}^n$ is articulated as follows:

$$p_{\hat{y}_{pf}^n|\varphi_p^n,\phi_p^n,\psi_p}\left(\hat{y}_{pf}^n \mid \varphi_p^n, \phi_p^n, \psi_p\right) \sim \mathcal{N}\left(\mu_n, \sigma_n^2\right)$$
$$\text{with} \quad \mu_n, \sigma_n^2 = g_{ep}^n(\varphi_p^n, \phi_p^n, \psi_p), \tag{6}$$

where $g_{ep}^n(\cdot)$ refers to EPM$_n$.

In contrast to VoxelDNN, which operates on voxel-level dependencies, our method embraces slice-level dependency, significantly enhancing the decoding speed. Moreover, for the first time, we incorporate side information from different modalities into the entropy model, enhancing its capability to predict symbolic probabilities more accurately. The detailed structures of additional sub-modules within CCEM will be presented in the Supplementary Materials.

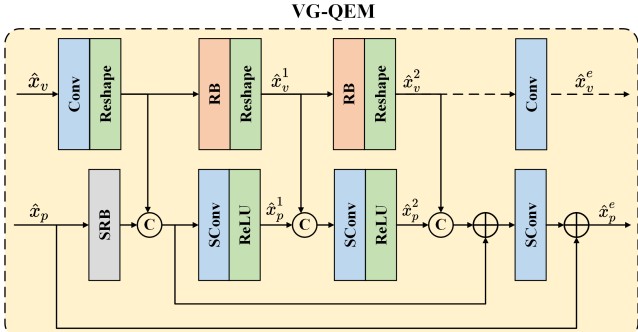

**Figure 4: The architecture of VG-QEM. It is important to highlight that the dashed line represents that the enhanced output image $\hat{x}_v^e$ is only employed in the calculation of the view image distortion loss term.**

## 3.4 View-Guided Quality Enhancement Module

At the end of the decoder, we feed the output point cloud $\hat{x}_p$ and view image $\hat{x}_v$ into VG-QEM to recover the local texture and geometric structure of the point cloud, aiming for achieving superior reconstruction quality. VG-QEM is coupled with the decoder, enabling an end-to-end training strategy that synergistically optimizes both the encoder-decoder network and VG-QEM. In VG-QEM, $\hat{x}_p$ and $\hat{x}_v$ are first extracted for features separately. Subsequently, $\hat{x}_v$ undergoes a sequence of dimensional transformation modules, inclusive of residual blocks and reshape operators, to enrich local details, resulting in $\hat{x}_v^i$. Here, $i$ represents the iteration count of dimensional transformations applied to $\hat{x}_v$, with $i \in \{1, 2\}$. In each level of operation, $\hat{x}_v^i$ is concatenated on the channel dimension with the aggregated point cloud features $\hat{x}_p^i$ through the refine module comprising sparse convolution and ReLU. This operation facilitates the coordinate reconstruction of the point cloud at edge details. Finally, through residual connections, we obtain the enhanced output point cloud $\hat{x}_p^e$.

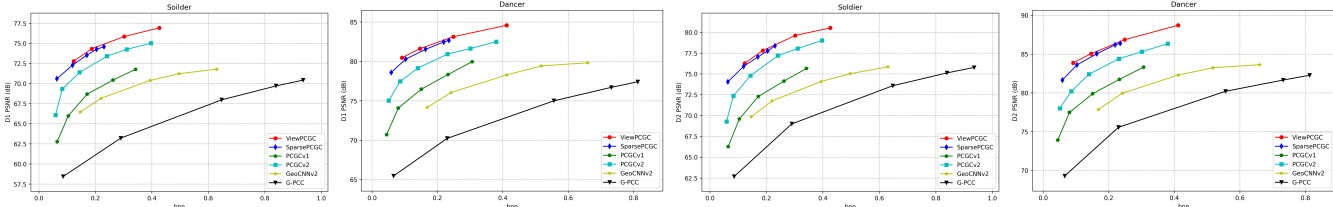

**Figure 5: Rate-distortion curves for performance comparision. From left to right are D1 PSNR of *Soilder*, D1 PSNR of *Dancer*, D2 PSNR of *Soilder*, and D2 PSNR of *Dancer*.**

## 3.5 Loss Function

In the training stage, we define the loss function $L$ in the following manner:

$$L = R_p + R_v + \alpha D_p + \beta D_v, \tag{7}$$

where $R_p$ and $D_p$ denote the rate loss and Binary CrossEntropy (BCE) loss of the point cloud, respectively. Similarly, $R_v$ and $D_v$ stand for the rate loss and Mean Squared Error (MSE) loss of the view image. $\alpha, \beta$ are the hyper-parameters of the optimization paradigm. Furthermore, $R_p, R_v, D_p, D_v$ can be expressed as:

$$
\begin{aligned}
R_p &= \mathbb{E}\left[-\log_2 p_{\hat{y}_{pf}|\varphi_p^a, \varphi_p^n, \phi_p^a, \phi_p^a, \psi_p}\left(\hat{y}_{pf} \mid \varphi_p^a, \varphi_p^n, \phi_p^a, \phi_p^a, \psi_p\right)\right] \\
&+ \mathbb{E}\left[-\log_2 p_{\hat{z}_p}\left(\hat{z}_p\right)\right], \\
R_v &= \mathbb{E}\left[-\log_2 p_{\hat{y}_v|\varphi_v}\left(\hat{y}_v \mid \varphi_v\right)\right] + \mathbb{E}\left[-\log_2 p_{\hat{z}_v}\left(\hat{z}_v\right)\right], \\
D_p &= -\frac{1}{N_p M}\sum_i^{N_p}\sum_j^M \left(x_i^j \log\left(p_i^j\right) + \left(1 - x_i^j\right)\log\left(1 - p_i^j\right)\right), \\
D_v &= \frac{1}{N_v}\|x_v - \hat{x}_v^e\|_2^2,
\end{aligned} \tag{8}
$$

where $N_p$ and $N_v$ represent the symbol length of $\hat{y}_p$ and $\hat{y}_v$, respectively. $M$ signifies the number of scales utilized within the decoder. $x_i^j$ stands for the status of a voxel, indicating its occupation or vacancy, and $p_i^j$ reflects the likelihood of the voxel being occupied.

It is crucial to recognize that although our primary concern in this research is the visual quality of the point cloud, the significance of the view image in reconstructing the point cloud has been previously highlighted in modules like MRRM, CCEM, VG-QEM. Consequently, we have included MSE loss for the view image in the loss function, to some extent preserving the reconstruction quality of the image. Moreover, in order to reduce the bit overhead associated with transmitting images, we aim to minimize the bit rate allocated to images during training. The dependency relationship between image reconstruction and point cloud reconstruction will be elaborated upon in more detail during our ablation study.

## 4 EXPERIMENTS

### 4.1 Datasets

*4.1.1 Training Dataset.* ShapeNet [5] is a large-scale dataset of 3D objects and stands as one of the most comprehensive datasets with rich annotations of 3D objects to date. The dataset covers several categories such as furniture, vehicles, and buildings. In our study, we randomly select 25,000 3D models from ShapeNet and divide the data into training and validation sets in a 9:1 ratio. These 3D

models are processed through sampling and quantization to obtain object point clouds. To generate the view image corresponding to each point cloud, we employ the virtual camera to capture the point clouds from the fixed perspective, thus obtaining training pairs.

*4.1.2 Test Dataset.* The 8iVFB dataset [11] specializes in highly realistic human three-dimensional scanning data. The MVUB dataset [6], provided by Microsoft, concentrates on the three-dimensional voxel representation of the upper body. Owlii, a company renowned for its advanced 3D scanning and reconstruction technology, offers a dataset [49] that primarily encompasses high-accuracy 3D human models and movements. These datasets are well-suited to meet the point cloud geometry compression requirements of MPEG standard [37] and JPEG standard [32]. Consequently, for our evaluation, we meticulously select a total of twelve point clouds from 8iVFB, MVUB, and Owlii datasets.

## 4.2 Experimental Details

*4.2.1 Training Strategy.* We train the entire network jointly. View-PCGC is implemented using PyTorch with CUDA support. During the training process, we utilize the Adam optimizer [22]. The learning rate is initialized at $1e^{-4}$ and gradually decreases as the model updates, eventually reaching $1e^{-5}$. The batch size is set to 4. Training is conducted on an NVIDIA RTX 3090, with each model undergoing approximately 40 training epochs. The point clouds in the input training pairs are quantized to 7 bits, and view images are cropped to $256 \times 256$ to facilitate model inference. The hyper-parameter $\beta$ is set to 200. Meanwhile, $\alpha$ is adjusted over a range from 0.5 to 10. The selection details of the hyper-parameter $\beta$ will be discussed in the Supplementary Materials.

*4.2.2 Evaluation Metric.* We adopt point-to-point PSNR (D1 PSNR) and point-to-plane PSNR (D2 PSNR) [20] as evaluation metrics, which reflect the fidelity of the spatial position of points in the point cloud and the fidelity of the geometric structure of the point cloud, respectively. Furthermore, Bjontegaard delta rate (BD-Rate) [4] is utilized to obtain a quantitative rate-distortion performance.

*4.2.3 Baseline.* We compare our method with several superior deep learning-based point cloud geometry compression methods, including GeoCNNv2 [36], PCGCv1 [45], PCGCv2 [44], and SparsePCGC [43]. Additionally, the classical point cloud encoding tool, G-PCC [28], is also incorporated into the comparison. The reference implementation for G-PCC is TMC13v23.

**Table 1: BD-Rate gains measured utilizing D1-PSNR and D2-PSNR metrics for ViewPCGC against G-PCC, GeoCNNv2, PCGCv1, PCGCv2, SparsePCGC in three test dataset.**

| Dataset | Point Cloud | G-PCC | | GeoCNNv2 | | PCGCv1 | | PCGCv2 | | SparsePCGC | |
|---|---|---|---|---|---|---|---|---|---|---|---|
| | | D1 | D2 | D1 | D2 | D1 | D2 | D1 | D2 | D1 | D2 |
| 8iVFB | longdress | -95.13% | -89.58% | -82.67% | -79.49% | -74.12% | -71.88% | -44.98% | -43.10% | -10.06% | -7.85% |
| | loot | -93.56% | -90.47% | -79.31% | -76.29% | -71.06% | -66.32% | -42.20% | -39.74% | -9.81% | -7.34% |
| | redandblack | -93.88% | -89.37% | 80.28% | -77.15% | -70.44% | -66.26% | -44.17% | -40.93% | -10.45% | -8.79% |
| | soldier | -92.24% | -88.31% | -81.24% | -77.73% | -72.68% | -69.32% | -40.49% | -38.40% | -10.73% | -8.24% |
| | **Average** | **-93.70%** | **-89.43%** | **-80.87%** | **-77.66%** | **-72.07%** | **-68.44%** | **-42.96%** | **-40.54%** | **-10.26%** | **-8.05%** |
| MVUB | andrew | -90.28% | -85.41% | -72.63% | -66.84% | -66.79% | -62.46% | -38.44% | -36.29% | -8.37% | -6.08% |
| | david | -89.94% | -86.17% | -74.29% | -69.98% | -68.47% | -63.56% | -39.18% | -36.17% | -8.78% | -7.14% |
| | phil | -91.18% | -88.65% | -76.71% | -72.18% | -69.03% | -65.58% | -40.74% | -38.01% | -9.54% | -7.62% |
| | sarah | -90.68% | -87.39% | -73.38% | -68.17% | -66.47% | -62.91% | -40.08% | -37.78% | -9.01% | -6.89% |
| | **Average** | **-90.52%** | **-86.90%** | **-74.25%** | **-69.29%** | **-67.69%** | **-63.62%** | **-39.61%** | **-37.06%** | **-8.92%** | **-6.93%** |
| Owlii | basketball_player | -95.37% | -89.16% | -85.74% | -80.92% | -77.61% | -72.42% | -48.32% | -44.64% | -12.01% | -10.42% |
| | dancer | -93.74% | -89.25% | -83.86% | -82.03% | -73.85% | -71.24% | -45.80% | -43.72% | -11.71% | -9.22% |
| | exercise | -96.43% | -90.27% | -85.29% | -83.31% | -76.38% | -74.24% | -47.92% | -45.78% | -11.87% | -9.86% |
| | model | -95.89% | -89.73% | -82.16% | -78.94% | -74.42% | -69.87% | -44.66% | -40.93% | -13.25% | -10.78% |
| | **Average** | **-95.35%** | **-89.60%** | **-84.26%** | **-81.30%** | **-75.56%** | **-71.94%** | **-46.67%** | **-43.76%** | **-12.21%** | **-10.07%** |
| | **Total average** | **-93.19%** | **-88.64%** | **-79.79%** | **-76.08%** | **-71.77%** | **-68.00%** | **-43.08%** | **-40.45%** | **-10.46%** | **-8.35%** |

## 4.3 Experiment Results

*4.3.1 Quantitative Results.* Table 1 presents the RD performance comparison of ViewPCGC on 8iVFB, MVUB, and Owlii datasets against other competitive methods. Overall, our proposed method achieves an average improvement over the existing state-of-the-art model by more than 10% and 8% on D1-PSNR and D2-PSNR metrics, respectively. Specifically, our method performs best on the Owlii dataset, attributed to its vast number of points, which further reduces the image bitstream expense. The aspect will be further discussed in the ablation study. Moreover, we plot the RD curves for *Soldier* and *Dancer* to further visualize the performance gap between various methods, as shown in Fig. 5. It illustrates that ViewPCGC is capable of better restoring the point cloud structure and improving the signal fidelity of the point cloud as the bitrate increases.

**Table 2: Average encoding time and decoding time comparision of different competitive methods.**

| Model | 8iVFB | MVUB | Owlii | Average |
|---|---|---|---|---|
| | Enc/Dec(s) | Enc/Dec(s) | Enc/Dec(s) | Enc/Dec(s) |
| G-PCC | 8.37/4.69 | 5.45/3.23 | 17.28/11.36 | **10.36/6.42** |
| PCGCv2 | 0.50/0.88 | 0.33/0.43 | 0.94/2.02 | **0.59/1.11** |
| SparsePCGC | 1.37/3.64 | 0.98/2.79 | 2.54/5.61 | **1.63/4.01** |
| ViewPCGC | 1.20/1.51 | 0.80/1.76 | 1.99/3.19 | **1.33/2.15** |

*4.3.2 Qualitative Results.* To clearly demonstrate the subjective visual effect of each model, we visualize the point cloud compressed by ViewPCGC and several competitive methods, as shown in Fig. 6.

For fairness, we attempt to compress all models at the same bitrate, with their actual bitrates arranged from left to right in the order of 0.27, 0.26, 0.22, 0.21, 0.18, as presented in the figure. We zoom in on the local details of *Soldier* (such as textures on clothing) to more intuitively showcase our reconstruction results. It is evident that G-PCC exhibits noticeable block effects during reconstruction, while methods like SparsePCGC suffer from varying degrees of white speckles, indicating geometric distortions in those areas. Our method, under the premise of using a lower bitrate, achieves the closest resemblance to the original point cloud texture, highlighting the superior subjective visual quality of ViewPCGC.

*4.3.3 Running Time.* We compare the encoding and decoding time of ViewPCGC with several competitive methods, and the test results are shown in Table 2. The hardware conditions for the test include a workstation equipped with an Intel(R) Xeon(R) Silver 4210R CPU and an NVIDIA GeForce RTX 3090 GPU. During the tests, we calculate the average encoding and decoding time across multiple bitrate points, from high to low. As seen in Table 2, compared to G-PCC, our method is much faster, especially on large point cloud datasets like Owlii. Compared to SparsePCGC, our method significantly reduces encoding and decoding time due to the more streamlined process of our model in handling point clouds across various scales. Although there is a certain gap in encoding and decoding time compared to PCGCv2, our method shows a noticeable improvement in RD performance. A pivotal aspect of ViewPCGC, enhancing its swift encoding and decoding performance, is the implementation of linear residual attention, significantly streamlining computational efficiency. We will describe more details about linear residual attention in the Supplementary Materials.

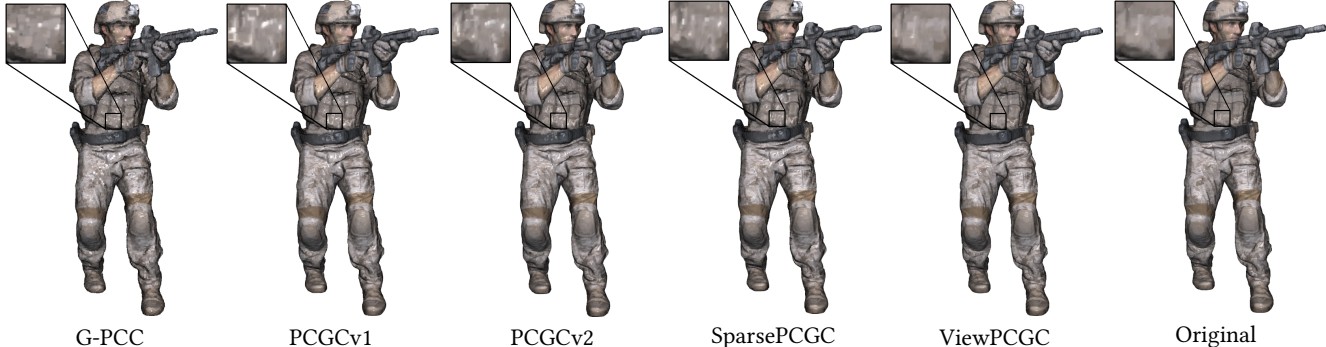

G-PCC       PCGCv1      PCGCv2      SparsePCGC     ViewPCGC     Original

**Figure 6: Visual quality comparison in *Soldier* for different competitive methods.**

## 4.4 Ablation Study and Analysis

*4.4.1 Case 1: Effectiveness of proposed modules.* To evaluate the effectiveness of the proposed modules, we conduct the ablation study to verify the contribution of each module, with the specific configurations detailed in Table 3. "✓" signifies that the proposed module is utilized within the baseline model. BD-Rate reflects the performance gap between the ablation model and the model proposed in this paper. Notably, when the CCEM is omitted, we employ a factorized entropy model [2] solely on the point cloud as an alternative. Table 3 illustrates the performance gap among models with various configurations, using ViewPCGC as the baseline model. The experimental results underscore the necessity of our proposed modules, among which CCEM contributes most significantly to performance enhancement, which can be attributed to the efficient utilization of modality contexts and accurate symbol probability estimation.

**Table 3: Ablation study of Case 1.**

| MRRM | CCEM | VG-QEM | BD-Rate(%) |
|:---:|:---:|:---:|:---:|
| ✓ | | | 18.74 |
| | ✓ | | 15.18 |
| | | ✓ | 20.77 |
| ✓ | ✓ | | 6.37 |
| ✓ | | ✓ | 12.65 |
| | ✓ | ✓ | 8.41 |

*4.4.2 Case 2: Effectiveness of suitable image resolution.* To determine the appropriate resolution for the view image, we perform the ablation study with the view image at several resolutions. Using the "256×256" resolution from our scheme as the baseline, Table 4 shows the bit overhead for utilizing the image at other resolutions as auxiliary information (left) and the average performance gap compared to the baseline (right) under D1-PSNR metric across three datasets. The bit overhead $R_o$ for the point cloud can be defined as:

$$R_o = \frac{R_v \times H \times W}{N}, \tag{9}$$

where $R_v$ denotes bpp for compressing the view image. $H$ and $W$ stand for height and weight of the view image, respectively. $N$ is the point number of the point cloud. It is clearly observed that

with the increase in image resolution, the rapid growth in image bitrate overhead leads to a sharp decline in overall performance. Conversely, when the image resolution is too low, the inability to extract sufficient modality information becomes the bottleneck, limiting performance improvement.

**Table 4: Ablation study of Case 2.**

| Resolution | 8iVFB | MVUB | Owlii |
|:---:|:---:|:---:|:---:|
| 192×192 | 0.0055/4.17% | 0.0155/1.28% | 0.0018/5.93% |
| 256×256 | 0.0088 | 0.0248 | 0.0028 |
| 512×512 | 0.0301/10.98% | 0.0849/21.72% | 0.0098/3.46% |
| 768×768 | 0.0651/14.62% | 0.1837/42.65% | 0.0213/10.05% |
| 1024×1024 | 0.1101/30.78% | 0.3102/57.39% | 0.0360/13.70% |

*4.4.3 Case 3: Effectiveness of joint optimization paradigm.* To validate the effectiveness of the joint optimization paradigm, we establish control groups for comparative analysis: A1 focuses solely on optimizing the bitrate and distortion of the point cloud; A2 extends this optimization to include the image's bitrate alongside the point cloud's bitrate and distortion; and A3, our adopted method, optimizes both the bitrate and distortion for the point cloud and the image. Results across three datasets using D1-PSNR metric reveal that A3 surpassed A1 by 5.72% and A2 by 8.14%. The results imply that while our goal is to reduce the image's bitrate overhead as much as possible, the quality of the image can not be completely overlooked, as it would affect the utilization of modality information from the image by the point cloud.

## 5 CONCLUSION

In this paper, we propose a novel view-guided learned point cloud geometry compression framework, namely ViewPCGC. To the best of our knowledge, it is also the first work to harness modality information from the projected view in learned object point cloud geometry compression. We employ MRRM to eliminate modality redundancy, CCEM to capture inter-modality context dependencies to assist in symbol probability estimation, and VG-QEM to restore local details of point cloud geometry and maintain spatial structure. The outstanding experiment results unveil that a new perspective of object point cloud geometry compression has been established by emphasizing the critical role of view image information.

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
