# OpenReview forum: "ViewPCGC: View-Guided Learned Point Cloud Geometry Compression"
_acmmm.org/ACMMM/2024/Conference — MM2024 Poster_

### Official Review · Reviewer_urKV · 2024-05-08

**Rating:** 4
**Confidence:** 1

**Summary:**

The core concept of the ViewPCGC framework is to utilize view information to assist in point cloud compression. By leveraging self-attention and cross-attention mechanisms based on sparse convolution, the framework aims to eliminate redundancy between modalities and enhance compression efficiency by incorporating side information from the view image into the encoding process. Additionally, the framework includes a View-Guided Quality Enhancement Module (VG-QEM) in the decoder to leverage the structural information of the view image for restoring local details in the point cloud, thereby improving the reconstruction quality.

**Strengths:**

- The paper introduces a novel view-guided learned point cloud geometry compression framework, which leverages modality information from view images to enhance compression efficiency. This innovative approach sets it apart from traditional compression methods.
- The framework effectively incorporates self-attention and cross-attention mechanisms based on sparse convolution to eliminate redundancy between modalities, leading to improved compression ratios and performance gains.

**Limitations:**

I apologize for my limited understanding of this field, but I think some additional clarifications are required in this paper.

- Table 1 presents several point cloud models for comparison, aligning with previous research practices. However, the lack of explanations within the table content might confuse readers, particularly those new to the field. Is the score of viewPCGC in D1-PSNR considered 100%? How is the average calculated? Providing such details would enhance reader comprehension.
- Why is PCGCv1 not included in Table 2 for comparison?
- To improve clarity, arrows could be added to Table 3 to indicate that a lower BD-rate is preferable. Additionally, in the ablation study case 1, explaining how the gap is calculated and which dataset is used would be beneficial.

Miscellaneous:

- The text in Figure 5 should be enlarged as it is currently difficult to read.

**Suitability:**

3

---

### Official Review · Reviewer_EBaM · 2024-05-11

**Rating:** 4
**Confidence:** 3

**Summary:**

Point clouds typically have large volumes, making them inconvenient to transmit over networks. Therefore, compression encoding and decoding are very important. The ViewPCGC point cloud geometry compression framework proposed in this paper introduces image information to optimize the compression and detail reconstruction of point clouds, improving compression efficiency and enhancing compression quality. In addition, the authors conducted extensive ablation experiments on each module, and the overall structure of the paper is complete.

**Strengths:**

see the summary part

**Limitations:**

1.	The reviewer suggests that the authors provide a detailed explanation of how the sparse self-attention (SSA) module and sparse convolution are implemented, as the current description is not specific enough to allow for reproducibility.
2.	Point cloud compression technology is a hot topic in academic research; the authors should compare their work with the state-of-the-art papers.
Rui Song, Chunyang Fu, Shan Liu, Ge Li; “Efficient Hierarchical Entropy Model for Learned Point Cloud Compression”  CVPR, 2023, pp. 14368-14377
Junteng Zhang et al. “YOGA: Yet Another Geometry-based Point Cloud Compressor”, ACM MM 2023
3.	The author mentioned that “Compared to point clouds, the bit cost of the view image during transmission is almost negligible”. The authors should provide detailed data to substantiate this viewpoint, such as discussions in Section 4.4.2 on the impact of the file size of images at different resolutions on transmission.
4.	As the authors reviewed in Section 4.3.1, the model proposed in this paper performs best on the Owlii dataset, with the author's explanation being that the larger number of points reduces the image bitstream expense. The reviewer believes that a specific analysis should be conducted on the impact of the density and sparsity of point clouds on the algorithm's performance.
5.	If the surface of a point cloud is uneven and bumpy, it can be difficult for the image to cover the surface texture comprehensively. The question remains whether the model can still be effective under such circumstances. The author should conduct more experiments to test the model's scalability.

**Suitability:**

3

---

### Official Review · Reviewer_KpEZ · 2024-05-13

**Rating:** 4
**Confidence:** 3

**Summary:**

In this paper, the authors propose ViewPCGC, a learning-based compression model for point clouds. ViewPCGC suggests to incorporate a 2D view image into the point cloud encoding and decoding process. Specifically, a model redundancy reduction module (MRRM) is designed to eliminate redundancy between modalities. In the entropy model, the conditional checkboard entropy model (CCEM) is proposed to refine the probability estimation for point cloud compression. In the decoder, a view-guided quality enhancement module (VG-QEM) is used to improve the output point cloud quality with the reconstructed view image. The experimental results show that ViewPCGC outperforms other PCGC methods on various datasets.

**Strengths:**

The draft is well-organized and is easy to follow. Employing the view image to improve the compression performance of point clouds is an interesting idea. The experimental results justify the designs, where the proposed method outperforms other learning-based compression methods on various data.

**Limitations:**

As the major contribution of this paper is that a view image is used in the learning-based compression model, the authors do not illustrate how they generate the view image. I only find a sentence 'To generate the view image corresponding to each point cloud, we employ the virtual camera to capture the point clouds from the fixed perspective, ...'. What is the fixed perspective to capture the view image and why use such a fixed perspective is not discussed. I am also wondering if the view image has colors or not. Since the latent features of the view image are generated and compressed, the authors are also encouraged to also report its size in the compression domain rather than only saying '... the bit cost of the view image during transmission is almost negligible.'

Since the details of generating view images are missing, my major concern is that this paper fails to illustrate why incorporating a view image could improve the performance of a learning-based compression model that uses sparse convolution. Does the view image provide additional information? If yes, what is the additional information and why don't the authors use multiple view images from different angles? If no, what is the reason for the superior performance by using the view image?

**Suitability:**

3

---

### Official Review · Reviewer_xPWF · 2024-05-24

**Rating:** 3
**Confidence:** 2

**Summary:**

This document introduces ViewPCGC, a novel view-guided learned point cloud geometry compression framework, which leverages self-attention and cross-modal attention mechanisms based on sparse convolution to efficiently compress point cloud data. By utilizing the contour information from view images, ViewPCGC effectively reduces redundancy between modalities, enhances the accuracy of entropy estimation, and improves detail reconstruction in point clouds. Demonstrated through extensive experimentation, ViewPCGC achieves superior performance compared to existing methods, offering significant improvements in terms of compression efficiency and quality of reconstructed point clouds.

**Strengths:**

**StrengthsStrengths**
1. **Innovative Fusion Framework:** ViewPCGC utilizes view-guided learning for point cloud geometry compression, leveraging view information and cross-modal interactions to enhance compression efficiency. This method integrates self-attention and cross-attention mechanisms based on sparse convolution, effectively handling modal redundancies.
2. **Enhanced Compression Efficiency:** The method uses the Modality Redundancy Removal Module (MRRM) and the Conditional Checkboard Entropy Model (CCEM), which significantly enhance the accuracy of probability density function estimation for point cloud geometry. These innovations lead to notable performance improvements over existing methods.
3. **Quality Enhancement:** The View-Guided Quality Enhancement Module (VG-QEM) in the decoder uses the contour information of the point cloud from the view image to reconstruct details more accurately, preserving the geometric integrity and detail of the point cloud.
4. **Robust Experimental Validation:** Extensive experiments demonstrate superior performance compared to state-of-the-art point cloud geometry compression methods, with significant gains in standard metrics like D1-PSNR.

**Limitations:**

**Limitations:**
1. **Complexity and Resource Intensity:** The architecture, while effective, is complex, potentially requiring significant computational resources for processing, especially in real-time applications. This may limit its practical deployment in resource-constrained environments.
2. **Dependency on High-Quality View Images:** The effectiveness of the method is partly dependent on the quality of the view images used for guiding the compression. Poor quality or inadequately aligned view images could potentially degrade the performance of the system.
3. **Limited Research on Modality Discrepancies:** While the method effectively handles modality redundancies through sophisticated modules, the inherent discrepancies and alignment issues between different modalities (e.g., discrepancies between point cloud data and 2D view images) are not thoroughly addressed, which could affect the overall compression quality under varying conditions.

**Suitability:**

3

---

### Meta-Review · Area_Chair_HzBe · 2024-07-01

**Recommendation:** Accept (Poster)
**Confidence:** 5

**Metareview:**

This paper initially got mixed reviews. The authors' rebuttal addressed all reviewers' concerns, and all reviews reached a consensus on the acceptance of the paper.